# Association of polypharmacy with fall-related fractures in older Taiwanese people: age- and gender-specific analyses

Hsueh-Hsing Pan,[1,2] Chung-Yi Li,[3,4] Tzeng-Ji Chen,[5] Tung-Ping Su,[6] Kwua-Yun Wang[2,7,8]

T-P Su and K-Y Wang contributed equally.

For numbered affiliations see end of article.

**Correspondence to**
Professor Kwua-Yun Wang;
w6688@mail.ndmctsgh.edu.tw

## ABSTRACT

**Objective:** To elucidate the associations between polypharmacy and age- and gender-specific risks of admission for fall-related fractures.

**Design:** Nested case–control study.

**Setting:** This analysis was randomly selected from all elderly beneficiaries in 2007–2008, and represents some 30% of the whole older insurers using Taiwan's National Health Insurance Research Database.

**Participants:** We identified 5933 cases newly admitted for fall-related fractures during 2007–2008, and 29 665 random controls free from fracture.

**Primary and secondary outcome measures:** Polypharmacy was defined as the use of fall-related drugs of four or more categories of medications and prescribed related to fall within a 1-year period. Logistic regression models were employed to estimate the ORs and related 95% CIs. The interaction of polypharmacy with age and sex was assessed separately.

**Results:** Compared with those who consumed no category of medication, older people who consumed 1, 2, 3 and ≥4 categories of medications were all at significantly increased odds of developing fall-related fractures, with a significant dose–gradient pattern (β=0.7953; p for trend <0.0001). There were significant interactions between polypharmacy and age, but no significant interactions between polypharmacy and gender. The dose–gradient relationship between number of medications category and risk of fall-related fractures was more obvious in women than in men (β=0.1962 vs β=0.1873). Additionally, it was most evident in older people aged 75–84 years (β=0.2338).

**Conclusions:** This population-based study in Taiwan confirms the link between polypharmacy and increased risk of fall-related fractures in older people; and highlights that elderly women and older people aged 75–84 years will be the targeted participants for further prevention from fall-related fractures caused by polypharmacy.

## Strengths and limitations of this study

- This is a population-based study providing further evidence that polypharmacy may increase fall-related fracture risk in older people, especially in the middle older age groups.
- Clinicians and researchers must consider an algorithm that may effectively identify inappropriate polypharmacy prescriptions, and more attention should be paid to the middle older people to reduce the incidence of fall-related fractures caused by polypharmacy.
- We relied on the prescription records in the database, which did not provide information on how patients use medications in the real world, and the other potential risk factors of fall, such as environmental factors and living arrangement were not available from the database.

events threatening the independence of older people and occur around 30–40% of community-dwelling adults older than 65 years worldwide each year.[2] The prevalence of falls strongly increased with age, and falls were more common in women than in men.[3] Serious falls can cause fractures which lead to prolonged length of hospital stay for the elderly, worsening of quality of life, increased cost of healthcare and even death.[2 4–6]

The incidence rate of falls increases with age and peaks in those aged over 80 years.[3] Major risk factors of falls in the elderly include functional decline, musculoskeletal problems, neurological diseases, psychosocial characteristics and medications.[2 4 6] In terms of medications, polypharmacy usually increased the risk of falls higher than twofold,[2–4] and the main group of drugs are benzodiazepines, antidepressants, antipsychotics and antiepileptics.[7] Previous studies about the role of polypharmacy for falls in the elderly were focused on the association between hip fracture and the number of

## INTRODUCTION

Older people aged over 65 years represented 10.7% of the total population in 2010 in Taiwan.[1] Falls belong to the most common

medications used per day,[8] or were mainly descriptive and institution-based with limited sample size and rarely focused on fall-related fractures.[9] However, few studies have examined polypharmacy associated with fall-related drugs on the impact of age and gender difference.

In the current study, we designed a case–control study and aimed to assess the cumulative effect of the selected categories of medications on the risk of fall-related fractures to assess the association between polypharmacy and the risk of fall-related fractures in older people. In addition to this, the age-specific and gender-specific stratified analyses were also performed.

## METHODS
### Source of data
Data analysed in this study were retrieved from Taiwan's National Health Insurance (NHI) Research Database (NHIRD) supervised by the Bureau of National Health Insurance (BNHI).[10] Taiwan reformed health insurance programmes into the universal NHI system in 1995, and more than 99% of residents were enrolled in this programme in 2007.[10] The NHIRD contains beneficiary registration files and claim data for reimbursement, offers information and protects the privacy and confidentiality of all beneficiaries by encrypting the personal/institutional identification numbers.

This analysis was randomly selected from all elderly beneficiaries (1 102 828) in 2007–2008, and represents some 30% of the whole older insurers made by the Review Committee of the National Health Research Institutes (NHRI) which cooperates with the BNHI to manage the NHIRD. Access to the NHI data has been approved by the NHRI.

### Study design and selection of cases and controls
This is a nested case–control study design. Cases were older patients (≥65 years) hospitalised between 1 January 2007 and 31 December 2008, with a discharge diagnosis code of fracture (800–829) along with accidental fall (E880–E888) based on the International Classification of Diseases, 9th revision, Clinical Modifications (ICD-9-CM) External Cause of Injury Codes (9,10). A total of 15 054 patients were identified. In order to exclude the patients who fell during hospitalisation and those who encountered fractures due to causes other than fall, we limited the cases to those who also had the same diagnostic codes at the emergency visits on the same date of admission (n=6015). Additionally, to ensure that all cases encountered new episodes of fall-related fracture during the study period (ie, 2007–2008), we excluded those patients who had also encountered hospitalisations for fall or fracture, or who received outpatient services for fracture in 2006 (n=53). Patients hospitalised for transport accidents (ICD-9 code from E800 to E848) were further excluded (n=29), ending up with a total of 5933 eligible cases during the 2-year period (figure 1). The index date for each case was date of his/her first-time hospitalisation for fall-related facture during the study period.

The control group was randomly selected from the study cohort free from falls or fractures during these 2 years, with a case/control ratio of 1/5. In total, 29 665 control participants were selected in this study. The index date for each control participant was 1 January 2007.

### Information on fall-related risk factors
Demographic data including age on the index date, gender and urbanisation level of residential area were

**Figure 1** Flow chart of cases.

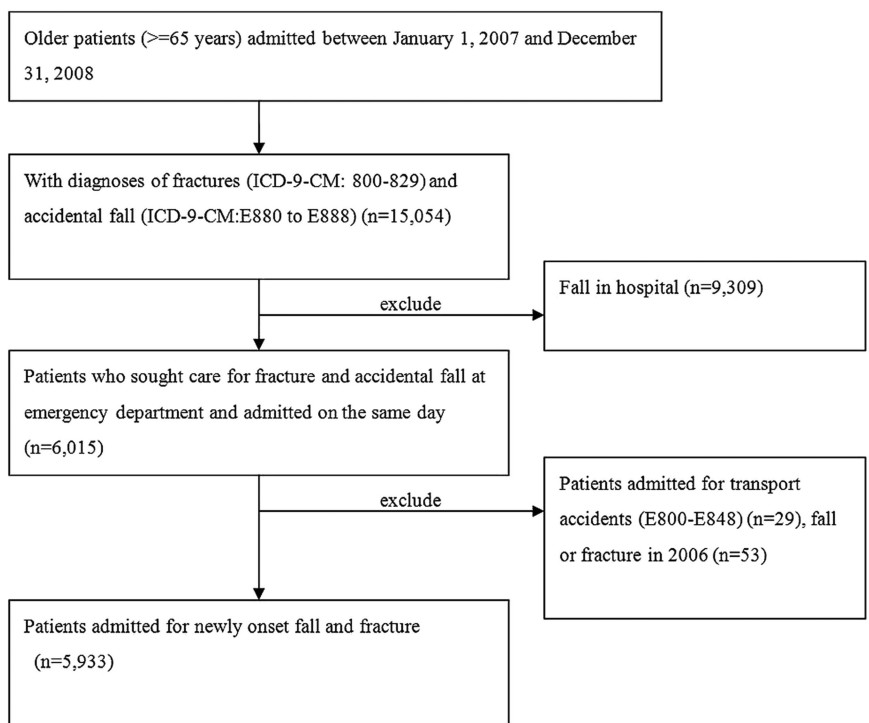

determined from the beneficiary registry. Age was divided into three groups: 65–74, 75–84 and ≥85 years.[8] The urbanisation level for each of the 365 townships in Taiwan was categorised (ie, urban, satellite city and rural) according to the National Statistics of Regional Standard Classification.[11] We retrieved information on the medical history for each participant during the 1-year period (ie, induction period) prior to the index date. The level of comorbidity was calculated using Charlson Comorbidity Index (CCI).[12]

The category of medications related to fall was counted and retrieved from medication history during the 1-year period and selected with WHO Anatomical Therapeutic Chemical (ATC) classification system including alimentary tract and metabolism, blood and blood forming organs, cardiovascular system, musculoskeletal system and nervous system.[7 9] More details are shown in table 1. Polypharmacy was defined as "the use of four or more categories of the selected fall-related medications."[3 13]

## Statistical analysis

Continuous variables were descriptively expressed as mean±SD and proportions for categorical variables. To determine whether polypharmacy is significantly associated with the risk of admission for fall-related fractures, logistic regression models are needed to calculate the crude and adjusted ORs and 95% CIs. In advance, the researchers investigated the association of age and gender, and polypharmacy with fall-related fracture admissions while controlling for other covariates. All statistical analyses were performed with SAS (V.9.2; SAS Institute, Cary, North Carolina). A p value <0.05 was considered statistically significant.

## RESULTS

### Baseline characteristics of the cases and controls

The mean age for cases and controls was 78.3±7.5 and 73.9±6.7 years, respectively. Female dominance was seen in cases, but not in controls. Cases were more likely than controls to live in rural areas (43.7% vs 36.6%), with a CCI score ≥3 (7.4% vs 5.0%) and polypharmacy (57.8% vs 41.5%; table 2).

### Polypharmacy and the risk factors of admission for fall-related fractures

Table 3 shows the OR of admission for fall-related fractures before and after adjusting the other variables. Compared with participants aged 65–74 years, the adjusted OR (AOR) of fall-related fractures for cases were 2.23 (95% CI 2.09 to 2.37) and 4.94 (95% CI 4.53 to 5.37) in those aged 75–84 and ≥85 years, respectively. There were 2.19-fold (95% CI 2.06 to 2.33) of admission for fall-related fractures in women compared with men. Older people living in rural areas were at significantly increased risk of admission (AOR=1.31, 95% CI 1.23 to 1.41). Compared with a CCI score of 0, the risk of admission for fall-related fractures increased with CCI score, especially in those with scores ≥3 (AOR=1.46, 95% CI 1.28 to 1.66). In addition, there was a 2.22-fold of admission among patients who used polypharmacy (≥4 categories of medications), as opposed to those used 0 category of medication.

### Age and gender specific for polypharmacy and admission for fall-related fractures

Table 4 shows AOR of admission for fall-related fractures in relation to category of medications according to age and gender. There was a significant interaction between category of medications and age. The significant dose–gradient relationship was noted for those aged 65–74 and 75–84 years, more obviously in those aged 75–84 years (β=0.2338; p for trend <0.0001). Compared with participants who used 0 category of medication, for participants who used polypharmacy, the AOR of admission for fall-related fractures increased to 2.46 and 2.40 for those aged 65–74 years and 75–84 years, respectively. Compared with those who used 0 category of medication, the risk of admission for fall-related fractures increased with the number of medication categories in men and women, and there was also a significant dose–gradient relationship. The AOR associated with polypharmacy was 2.16 for men, which was lower than that for women (AOR 2.23).

**Table 1** Medications related to fall calculated with WHO ATC classification system

| Category of medications related to fall | ATC index |
|---|---|
| Alimentary tract and metabolism | |
| Drugs for functional gastrointestinal disorders | A03 |
| Blood glucose lowering drugs and insulin | A10A, A10B |
| Vitamins | A11 |
| Blood and blood forming organs | |
| Antithrombotic agents | B01A |
| Cardiovascular system | |
| Cardiac glycosides and nitrates | C01A, C01DA |
| Antiarrhythmics | C01B |
| Antihypertensives | C02 |
| Diuretics | C03 |
| Musculoskeletal system | |
| NSAIDs | M01A |
| Nervous system | |
| Opioids | N02A |
| Antiepileptics | N03 |
| Antiparkinson drugs | N04 |
| Antipsychotics | N05A |
| Anxiolytics | N05B |
| Hypnotics and sedatives | N05C |
| Antidepressants | N06A |

ATC, Anatomical Therapeutic Chemical; NSAIDs, non-steroidal anti-inflammatory drugs.

**Table 2** Comparisons of sociodemographics and medications between cases and controls

| | Cases (n=5933) | | Controls (n=29 665) | | |
| --- | --- | --- | --- | --- | --- |
| | N | Per cent | N | Per cent | p Value |
| Age, mean±SD | 78.3 | ±7.5 | 73.9 | ±6.7 | <0.0001 |
| Age group, years | | | | | <0.0001 |
| 65–74 | 1956 | 33.0 | 17 082 | 57.6 | |
| 75–84 | 2671 | 45.0 | 10 270 | 34.6 | |
| ≥85 | 1306 | 22.0 | 2313 | 7.8 | |
| Gender | | | | | <0.0001 |
| Male | 1833 | 30.9 | 14 678 | 49.5 | |
| Female | 4100 | 69.1 | 14 987 | 50.5 | |
| Urbanisation | | | | | <0.0001 |
| Urban | 1930 | 32.5 | 10 934 | 36.9 | |
| Satellite | 1408 | 23.7 | 7884 | 26.6 | |
| Rural | 2595 | 43.7 | 10 847 | 36.6 | |
| CCI score | | | | | <0.0001 |
| 0 | 1869 | 31.5 | 12 398 | 41.8 | |
| 1 | 2595 | 43.7 | 11 982 | 40.4 | |
| 2 | 1030 | 17.4 | 3810 | 12.8 | |
| ≥3 | 439 | 7.4 | 1475 | 5.0 | |
| Category of medications | | | | | |
| 0 | 348 | 5.9 | 3464 | 11.7 | <0.0001 |
| 1 | 523 | 8.8 | 3984 | 13.4 | |
| 2 | 775 | 13.1 | 4913 | 16.6 | |
| 3 | 857 | 14.4 | 4995 | 16.8 | |
| ≥4 | 3430 | 57.8 | 12 309 | 41.5 | |

CCI, Charlson comorbidity index.

## DISCUSSION

This study used large population-based datasets to identify the relationship between polypharmacy and fall-related fractures in older people. The major findings were as follows: age, female, living in rural areas, comorbidities and category of medications were likely to have a higher risk of admission for fall-related fractures. An increasing category of medications related to fall is independently and positively associated with the risk of admission for fall-related fractures, especially in those younger older people but not obvious in those aged ≥85 years; the risk of admission for fall-related fractures increased with the category of medications in both men and women, and there was also a significant dose–gradient relationship.

Our findings were similar to many previous studies that risk factors for falls increase with age[4 14 15] and polypharmacy.[3 4 13 16 17] It is likely that older people with advanced age tend to have higher comorbidities, lack self-sufficiency, are frail and suffer from more severe forms of osteoporosis, leading to an increased risk of frequent falls and more common fractures.[18 19] Ziere et al[3] conducted a cross-sectional study of nearly 7000 individuals aged ≥55 years and noted that falls were more common in women than in men. A systematic review which focused on the risk factors for falls in older people also showed that women at an advanced age were more prone to falls.[20] Our study indicated that women were more likely to suffer from

fall-related fractures than men. A possible explanation was that Asian women were at a high risk of osteoporosis and had a higher incidence of hip fractures than men.[21] Older people living in rural areas were likely to have a higher risk of admission for fall-related fractures. Yiannakoulias et al[22] investigated the relationship between geographical locations and fall injuries and the study found that the surrounding rural regions and smaller communities had a more moderate fall incidence. A survey of falls in Taiwan indicated that most of the fall events took place outdoors, including streets sidewalks, farmlands and mountain areas; and those who experienced fall indoors suffered the episode mainly in the living room and bathroom.[23] A cross-sectional study found that urban residents had fewer medical diagnoses, better mobility, less pain and fewer depressive symptoms compared with rural residents.[24] Therefore, disadvantageous indoor and outdoor environments experienced by the older people from rural Taiwan might explain the urbanisation and fall relationship noted in our findings.

Huang et al[17] found a statistically significant interaction between total number of medications and falls by different age groups (<65, ≥65 years of age) in a multiethnic population of patients with type 2 diabetes. They found a significant increase in the risk of falls with 4–5 and 7 or more medications, and the HR were 1.45 and 2.31, respectively, compared with 0–1 prescription medications. In older patients, there was a monotonic rise in

**Table 3** Crude and adjusted ORs and 95% CIs of admission for fall-related fractures in relation to sociodemographics and medications

| | Crude estimate | | Adjusted estimate | |
|---|---|---|---|---|
| | OR | 95% CI | OR | 95% CI |
| **Age group, years** | | | | |
| 65–74 | 1.00 | Reference | 1.00 | Reference |
| 75–84 | 2.27 | 2.13 to 2.42 | 2.23 | 2.09 to 2.37 |
| ≥85 | 4.93 | 4.54 to 5.36 | 4.94 | 4.53 to 5.37 |
| | | β=0.7953; p for trend <0.0001 | | |
| **Gender** | | | | |
| Male | 1.00 | Reference | 1.00 | Reference |
| Female | 2.19 | 2.06 to 2.33 | 2.19 | 2.06 to 2.33 |
| **Urbanisation** | | | | |
| Urban | 1.00 | Reference | 1.00 | Reference |
| Satellite | 1.01 | 0.94 to 1.09 | 1.03 | 0.96 to 1.12 |
| Rural | 1.36 | 1.27 to 1.45 | 1.31 | 1.23 to 1.41 |
| | | β=0.1370; p for trend <0.0001 | | |
| **CCI** | | | | |
| 0 | 1.00 | Reference | 1.00 | Reference |
| 1 | 1.44 | 1.35 to 1.53 | 1.08 | 1.00 to 1.16 |
| 2 | 1.79 | 1.65 to 1.95 | 1.26 | 1.15 to 1.39 |
| ≥3 | 1.97 | 1.76 to 2.22 | 1.46 | 1.28 to 1.66 |
| | | β=0.1008; p for trend <0.0001 | | |
| **Category of medications** | | | | |
| 0 | 1.00 | Reference | 1.00 | Reference |
| 1 | 1.31 | 1.13 to 1.51 | 1.29 | 1.11 to 1.49 |
| 2 | 1.57 | 1.37 to 1.80 | 1.46 | 1.33 to 1.60 |
| 3 | 1.71 | 1.50 to 1.95 | 1.51 | 1.31 to 1.73 |
| ≥4 | 2.77 | 2.47 to 3.12 | 2.22 | 1.95 to 2.52 |
| | | β=0.1943; p for trend <0.0001 | | |

CCI, Charlson comorbidity index.

the risk of falls that became statistically significant at 6–7 (HR=1.28) or more than 7 medications (HR=1.39), compared with 0–1 prescription medications. Actually, accompanying deterioration in bone quality in older people may be the high-risk group for fall-related fractures.[25] [26] In short, older people with advanced age tend to have higher comorbidities so that the medication threshold at which fall risk begins to rise may be lower in older patients than younger patients. We divided a number of medications among different groups, and showed that there was an increased risk of admission for fall-related fractures in older people who had higher use of medications, with a significant dose–gradient relationship, which further substantiates the link between polypharmacy and risk of fall-related fracture. Consequently, regular reviews of older adults' medications, minimising the drugs unless those are really necessary, may be an appropriate and costless fall prevention strategy.

**Table 4** Covariate adjusted ORs and 95% CI of admission for fall-related fractures in relation to category of medications according to gender and age

| Category of medications | Gender | | | | Age (year) | | | | | |
|---|---|---|---|---|---|---|---|---|---|---|
| | Men | | Women | | 65–74 | | 75–84 | | ≥85 | |
| | AOR | 95% CI | AOR | 95% CI | AOR | 95% CI | AOR | 95% CI | AOR | 95% CI |
| 0 | 1.00 | Reference | 1.00 | Reference | 1.00 | Reference | 1.00 | Reference | 1.00 | Reference |
| 1 | 1.31 | 1.03 to 1.67 | 1.26 | 1.05 to 1.52 | 1.49 | 1.17 to 1.90 | 1.11 | 0.86 to 1.43 | 1.34 | 1.00 to 1.81 |
| 2 | 1.39 | 1.10 to 1.75 | 1.48 | 1.24 to 1.77 | 1.66 | 1.32 to 2.09 | 1.55 | 1.23 to 1.96 | 1.12 | 0.84 to 1.50 |
| 3 | 1.55 | 1.23 to 1.96 | 1.47 | 1.24 to 1.75 | 1.62 | 1.28 to 2.04 | 1.71 | 1.36 to 2.14 | 1.13 | 0.84 to 1.51 |
| ≥4 | 2.16 | 1.75 to 2.68 | 2.23 | 1.90 to 2.62 | 2.46 | 1.98 to 3.05 | 2.40 | 1.95 to 2.97 | 1.69 | 1.30 to 2.20 |
| | β=0.1873; p value for trend <0.0001 | | β=0.1962; p value for trend <0.0001 | | β=0.1962; p value for trend <0.0001 | | β=0.2338; p value for trend <0.0001 | | β=0.1168; p value for trend <0.0001 | |

AOR, adjusted OR; adjustment for age, urbanisation and Charlson index.

This study has the following strengths. First, it was population-based and included all eligible older patients in Taiwan. Therefore, the data are highly representative and allow little room for selection bias. Second, the longitudinal data and completed reimbursement claims allow for precise medication estimation. Third, the advantage of using insurance claim datasets in clinical research is easy access to the longitudinal records for a large sample of patients. Finally, we managed to identify, in a multivariate regression model, a number of demographic variables and CCI scores that are considered to be risk factors for fall-related fractures. This way may reduce the confounding, and point to the independent effect of polypharmacy on the risk of fall-related fractures in older patients.

Several limitations inherent with the use of administrative databases also need to be mentioned. First, we relied on the prescription records in the database, which did not provide information on how patients use medications in the real world. We also used a proxy indicator for medication use—medications purchased within the last year, not medication administration. It is possible that in a few instances medications were prescribed but not subsequently taken. Second, medication history was observed for only 1 year before the fall-related fractures. This may underestimate the medication-associated risk of fall-related fractures. Third, only falls related to hospitalisation and hospital treatment are presented. Falls which were treated in clinics, self-treated or that did not result in an injury are not included. Finally, the other potential risk factors of fall, such as environmental factors and living arrangement were not available from the NHI datasets, so that we were unable to take these potential confounders into consideration.

In summary, this population-based study provides further evidence that polypharmacy may increase fall-related fracture risk in older people, especially in the younger age groups. To further reduce the incidence of fall-related fractures caused by polypharmacy, clinicians and researchers must consider an algorithm that may effectively identify inappropriate polypharmacy prescriptions, and more attention should be paid to the younger older people.

**Author affiliations**
[1]Department of Nursing, Tri-Service General Hospital, Taipei, Taiwan
[2]School of Nursing, National Defense Medical Center, Taipei, Taiwan
[3]Graduate Institute and Department of Public Health, College of Medicine, National Cheng Kung University, Tainan, Taiwan
[4]Department of Public Health, College of Public Health, China Medical University, Taichung, Taiwan
[5]Department of Family Medicine, Taipei Veterans General Hospital, Taipei, Taiwan
[6]Department of Psychiatry, Taipei Veterans General Hospital, Taipei, Taiwan
[7]Department of Nursing, Taipei Veterans General Hospital, Taipei, Taiwan
[8]School of Nursing, National Yang-Ming University, Taipei, Taiwan

**Acknowledgements** The authors would like to thank Taipei Veterans General Hospital for sponsoring this study.

**Contributors** H-HP analysed the data and wrote the manuscript. C-YL, T-JC, T-PS and K-YW designed the study and revised the manuscript.

**Funding** This work was supported by a grant from Taipei Veterans General Hospital (V100C-211).

**Competing interests** None.

**Provenance and peer review** Not commissioned; externally peer reviewed.

**Data sharing statement** No additional data are available.

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
