## [Reviewer comments · BMJ Open]

Some articles will have been accepted based in part or entirely on reviews undertaken for other BMJ Group journals. These will be reproduced where possible.

ARTICLE DETAILS

TITLE (PROVISIONAL)	Association of polypharmacy with fall-related fractures in older Taiwanese people: age- and gender-specific analyses
AUTHORS	Wang, Kwua-Yun; Pan, Hsueh-Hsing; Li, Chung-Yi; Chen, Tzeng-Ji; Su, Tung-Ping

VERSION 1 - REVIEW

REVIEWER	Kuan-Fu Liao Department of Internal Medicine, Taichung Tzu Chi General Hospital, Taichung, Taiwan
REVIEW RETURNED	23-Jan-2014

GENERAL COMMENTS	Although the similar result was reported by using the same database (Polypharmacy correlates with increased risk for hip fracture in the elderly: a population-based study. Medicine (Baltimore) 2010; 89: 295-9. You have cited it), this is a very straight-forward study to explore the association between polypharmacy and fall-related fractures in older people. Also the used methodology (nested case-control study) and the analysis seems adequate. The discussion comprises the essential topics. I therefore recommend to accept this manuscript for publication.
---

REVIEWER	Taro Kojima Department of Geriatric Medicine, The University of Tokyo
REVIEW RETURNED	09-Feb-2014

GENERAL COMMENTS	– This paper could be acceptable, pending the appropriate responses to the comments. – The present study evaluated the association between polypharmacy and fall-related fractures in Taiwanese elderly people, and found that the number of medications is a risk of fall-related fractures in a stepwise manner. There are a few comments about the manuscript. Major points. 1. There are several drugs which are risks for fall. The authors counted the number of medications but did not assess the categories of drugs. Is the category of drugs available for this study? If so, is there any type of drugs which was associated with fall-related fractures? Is there a difference in prescription between 65-84
--

	years-old and 85 years or older? 2. Are there any specific comorbid conditions which were related to falls? (Components of CCI or other major diseases which are not included in components of CCI, e.g. hypertension, insomnia, and osteoarthritis.) Comments Using CCI for assessing disease burden of community-dwelling people is a little uncomfortable to this reviewer. It was originally established for assessing the prognosis of inpatients (J Chron Dis 40; 373-383, 1987), who are in severer condition. But there are several papers assessing CCI for non-inpatients, so it could be acceptable.
--	---

VERSION 1 – AUTHOR RESPONSE

Reviewer Name Kuan-Fu Liao

Institution and Country Department of Internal Medicine, Taichung Tzu Chi General Hospital, Taichung, Taiwan

Please state any competing interests or state 'None declared': None Declared

Although the similar result was reported by using the same database (Polypharmacy correlates with increased risk for hip fracture in the elderly: a population-based study. Medicine (Baltimore) 2010; 89: 295-9. You have cited it), this is a very straight-forward study to explore the association between polypharmacy and fall-related fractures in older people. Also the used methodology (nested case-control study) and the analysis seems adequate. The discussion comprises the essential topics. I therefore recommend to accept this manuscript for publication.

Authors' response:

Thank you very much. We have revised the above information on competing interests of the revised manuscript (page 21, last line).

Reviewer Name Taro Kojima

Institution and Country Department of Geriatric Medicine, The University of Tokyo

Please state any competing interests or state 'None declared': None declared

Authors' response:

Thank you very much. We have revised the above information on competing interests of the revised manuscript (page 21, last line).

Major points.

1. There are several drugs which are risks for fall. The authors counted the number of medications but did not assess the categories of drugs. Is the category of drugs available for this study? If so, is there any type of drugs which was associated with fall-related fractures? Is there a difference in prescription between 65-84 years-old and 85 years or older?

Authors' response:

Thank you very much for your kindness and valuable response. In respond to your recommendations, we have rechecked our data and observed additional findings. Based on these findings, we revised

our content and highlighted in blue. The category of drugs is also available for this study. We have analyzed the specific category drugs in association with fall-related fractures (please see the suppl. table 1 below). We found that there is a difference in the category of medications prescribed in patients with different ages, including blood glucose lowering drugs and insulin (A10A, A10B), antiparkinson drugs (N04), and antipsychotics (N05A) (data do not show). Furthermore, there is a difference in prescription between 65-84 years-old and 85 years or older (please see the suppl. table 2 below).

In this study, we aimed to assess the cumulative effect of these categories of medications on the risk of fall-related fractures, rather than to evaluate the association of specific category of medication with risk of fall-related fracture. This is mainly because that the above mentioned drug categories have been found to be associated with fall-related fracture in a number of studies (French et al., 2005; Hartikainen et al., 2007). Therefore, we counted the number of medication category, and associated the number of categories with the risk of fall-related fracture. To make it clear, we have provided the information on drug categories in Table 1 and introduction (page 6) of the revised manuscript.

French DD, Campbell R, Spehar A, et al. Outpatient medications and hip fractures in the US: a national veterans study. *Drugs Aging* 2005; 22: 877-85.

Hartikainen S, Lonroos E, Louhivuori K. Medication as a risk factor for falls: critical systematic review. *J Gerontol A Biol Sci Med Sci* 2007; 62: 1172-81.

2. Are there any specific comorbid conditions which were related to falls? (Components of CCI or other major diseases which are not included in components of CCI (e.g. hypertension, insomnia, and osteoarthritis.)

Comments

Using CCI for assessing disease burden of community-dwelling people is a little uncomfortable to this reviewer. It was originally established for assessing the prognosis of inpatients (*J Chron Dis* 40; 373-383, 1987), who are in severer condition. But there are several papers assessing CCI for non-inpatients, so it could be acceptable.

Authors' response:

Thank you very much for your important comments. Based on your comments, there are several papers assessing CCI for non-inpatients. In our study, we aimed to assess the cumulative effect associated with these categories of medications on the risk of fall-related fractures including antihypertensives, hypnotics and sedatives, and non-steroidal anti-inflammatory drugs (NSAIDs). These drugs also used to treat hypertension, insomnia, and osteoarthritis. If we adjusted these three diseases and drug categories in regression model at the same time, it may have collinearity.

VERSION 2 – REVIEW

REVIEWER	Taro Kojima Department of Geriatrics, The University of Tokyo
REVIEW RETURNED	28-Feb-2014

GENERAL COMMENTS	The authors have fully responded to the questions and are all acceptable.
---